# A faster and less costly alternative for RNA extraction of SARS-CoV-2 using proteinase k treatment followed by thermal shock

**Adolfo Marcelo Ñique[1]\*, Fiorella Coronado-Marquina[1], Jairo Andrés Mendez Rico[2], María Paquita García Mendoza[1], Nancy Rojas-Serrano[1], Paulo Vitor Marques Simas[1,3], Cesar Cabezas Sanchez[1], Jan Felix Drexler[4]**

1 National Institute of Health–INS (Instituto Nacional de Salud–INS), Lima, Peru, 2 Pan American Health Organization–PAHO, Washington, DC, United States of America, 3 Faculty of Veterinary Medicine, University of San Marcos (Universidad Nacional Mayor de San Marcos–UNMSM), Lima, Peru, 4 Charité–Universitätsmedizin Berlin, Berlin, Germany

\* adolfo.marcelo.n@gmail.com

**Citation:** Ñique AM, Coronado-Marquina F, Mendez Rico JA, García Mendoza MP, Rojas-Serrano N, Simas PVM, et al. (2021) A faster and less costly alternative for RNA extraction of SARS-CoV-2 using proteinase k treatment followed by thermal shock. PLoS ONE 16(3): e0248885. https://doi.org/10.1371/journal.pone.0248885

**Data Availability Statement:** All relevant data are within the manuscript and its Supporting Information files.

## Abstract

One of the biggest challenges during the pandemic has been obtaining and maintaining critical material to conduct the increasing demand for molecular tests. Sometimes, the lack of suppliers and the global shortage of these reagents, a consequence of the high demand, make it difficult to detect and diagnose patients with suspected SARS-CoV-2 infection, negatively impacting the control of virus spread. Many alternatives have enabled the continuous processing of samples and have presented a decrease in time and cost. These measures thus allow broad testing of the population and should be ideal for controlling the disease. In this sense, we compared the SARS-CoV-2 molecular detection effectiveness by Real time RT-PCR using two different protocols for RNA extraction. The experiments were conducted in the National Institute of Health (INS) from Peru. We compared Ct values average (experimental triplicate) results from two different targets, a viral and internal control. All samples were extracted in parallel using a commercial kit and our alternative protocol–samples submitted to proteinase K treatment (3 µg/µL, 56°C for 10 minutes) followed by thermal shock (98°C for 5 minutes followed by 4°C for 2 minutes); the agreement between results was 100% in the samples tested. In addition, we compared the COVID-19 positivity between six epidemiological weeks: the initial two in that the Real time RT-PCR reactions were conducted using RNA extracted by commercial kit, followed by two other using RNA obtained by our kit-free method, and the last two using kit once again; they did not differ significantly. We concluded that our in-house method is an easy, fast, and cost-effective alternative method for extracting RNA and conducing molecular diagnosis of COVID-19.

## Introduction

The Severe Acute Respiratory Syndrome 2 (SARS-CoV-2), the causative agent of Coronavirus Disease 2019 (COVID-19), has been classified as part of the Coronaviridae family. These

**Funding:** This study was supported by Nacional Institute of Health (NIH), state Institution associated whit Ministry of Health from Peru. We thank the Pan American Health Organization (PAHO) for providing primers, probes and referential controls and the initial protocol for RNA extraction using thermal shock.

**Competing interests:** The authors declare none competing interest.

viruses have a phospholipid envelope derived from the host membrane, containing the proteins that give it the crown shape [1]. Coronaviruses, as other enveloped viruses, are generally more vulnerable to heating because the heat denatures the envelope and makes them incapable of infection.

According to data available by World Health Organization (WHO), the SARS-CoV-2 can survive for up to 72 hours on plastic and stainless steel, less than 4 hours on copper, and less than 24 hours on cardboard [2]. However, the virus can be detected in environments after this time [3–5], indicating that environmental conditions like temperature, for example, can inactivate the viral particles but, in the meantime, is not able to completely degrade viral genetic material, making it possible to detect it.

SARS-CoV-2 can be inactivated at 60°C, 80°C, and 100°C for approximately 32.5, 3.7, and 0.5 minutes, respectively, and these times and temperatures held enough to reduce the infectivity in five logs [6]. These temperatures are also used during amplification by real time RT-PCR, indicating that the genetic material's quality is not compromised. Furthermore, using an internal control for Real time RT-PCR reaction can confirm the samples' integrity, decreasing the probability of false-negative results.

Based on this knowledge, physical methods like heating and specific chemical reagents like proteinase K are widely used in genetic material extraction protocols to optimize viral purification. Proteinase K is critical because it digests proteins eliminating contamination from nucleic acid preparations, in addition to inactivating the nucleases that could degrade DNA or RNA during purification.

With all of this, we established a protocol using proteinase K treatment of nasopharyngeal samples in Universal Transport Medium (UTM) followed by thermal shock for a faster and low-cost RNA extraction for COVID-19 diagnosis. We compared our new method's effectiveness based on Ct values obtained by Real time RT-PCR specific for SARS-CoV-2 using a constitutive human gene as an internal control.

## Material and methods

### Ethical considerations

The study was conducted at the INS, Peruvian Ministry of Health. The INS contains several reference laboratories that are in authority for epidemiological surveillance in human health in Peru. This article reported results obtained in the standardization process of an alternative method for RNA extraction and the retrospective study using this new protocol. All procedures were performed according to Pan American Health Organization (PAHO) and WHO guidelines, which is required for the implementation of a new protocol in any reference laboratory. The retrospective medical records and the archived samples were completely anonymous before accessing.

All patients voluntarily allowed the collection of a nasal and pharyngeal swab in search of a molecular diagnosis of COVID-19. All sample collection procedures were performed according to standards established by WHO, CDC, and mainly following the guidelines of the Declaration of Helsinki for the diagnosis of infectious diseases of the respiratory tract [7–9].

All experiments carried out for molecular diagnostics at the INS of Peru related to COVID-19 were authorized by a chieftain resolution under file number 00006918 provided in emergency decree 0064-2020-OGA/INS of April 7th, 2020 and the note information 0055–2020 "Plan de acción del Instituto Nacional de Salud para prevención, diagnóstico y control del COVID-19, en el marco del decreto supremo n° 008-2020-SA".

The retrospective study was not sent for evaluation by the ethics committee because it was included in INS's action plan from Peru. This supreme decree determined that the National

Reference Laboratory of Respiratory Viruses, local where the experiments presented in the paper were conducted, must be responsible for "Diagnóstico Molecular de Muestras de Casos Sospechosos de COVID-19".

## Experimental design

Our alternative protocol for RNA extraction was reasoned in the proteinase K properties and the SARS-CoV-2 biological characteristics. The low resistance to the temperature increases of both viruses and hosting cells and the enzyme's ability to destroy all proteins in the envelope, capsid, and the ones associated with viral RNA, provide the ideal mechanisms for stripping and solubilizing viral RNA.

The nasopharyngeal swabs were preserved in Universal Transport Medium (UTM), in which 100 μL was treated with 30 μg of proteinase K. The solution was incubated in a dry bath (56˚C, 10 minutes) followed by thermal shock (98˚C, 5 minutes; 4˚C, 2 minutes).

The SARS-CoV-2 molecular detection was conducted using real time RT-PCR multiplex to detect a viral-specific, RNA polymerase RNA dependent gene (*RdRp*) and a human transcript gene as internal control, Glyceraldehyde-3-Phosphate Dehydrogenase (*GAPDH*). The Ct values obtained in both channels were compared between protocols of the RNA purification and statistically analyzed. The acceptance criterion for validation of this new RNA extraction proceeding was 100% concordant compared to the extractions performed using a commercial kit. All these proceedings, from RNA extraction to amplification reactions, were performed by the same operator under the same conditions (environment, equipment, samples).

## Clinical samples

Seventy-eight samples were simultaneously submitted to both RNA extraction methods, using the commercial kit, according to manufacturers' instructions (Qiagen), and another using proteinase k treatment and thermal shock protocol.

After this standardization step and considering that there were no RNA extraction commercial kits available during the period between June 3rd and 25th at INS from Peru, all COVID-19 molecular diagnosis using Real time RT-PCR were made using RNA purified using proteinase k and thermal shock. To establish a real comparison of positivity obtained in this period, we compared the Real time RT-PCR results obtained in two complete epidemiological weeks (EW) before June 3rd (EW 21 and 22) and the other two after June 25th (EW 28 and 29). The EW 27 represented the quarantine ending on June 30th. The quarantine ending allowed more circulation of people, increasing the probability of virus transmission. Thus, this week was omitted because it could offer a bias in these results. The positivity between these six epidemiological weeks was compared, including 148,396 clinical samples evaluated by Real time RT-PCR from cases with suspected signs for COVID-19.

## SARS-CoV-2 detection by Real time RT-PCR multiplex

The real time RT-PCR multiplex used to detect SARS-CoV-2 in clinical samples swabs were standardized using primers and specific probes for SARS-CoV-2 *RdRp* gene [10] and included primers and probe for the constitutive human transcript for the enzyme *GAPDH* (Table 1) on the same reaction.

The reagents and conditions of Real time RT-PCR multiplex were summarized in Table 2. The plots for *RdRP* were evaluated in the Green channel (470 nm for excitation, 510 nm for detection, fluorescent FAM labeled probe). The plots for *GAPDH* were evaluated in the Orange channel (585 nm for excitation, 610 nm for detection, fluorescent ROX labeled probe). The results were analyzed considering the cycle threshold (Ct) values. Samples presenting Ct

**Table 1. Genes and probes used in the Real time RT-PCR multiplex to detect SARS-CoV-2 using *GAPDH* as an internal control.**

| Target gene | Primer / Probe | Sequence 5' → 3' |
|---|---|---|
| *RdRp* | *RdRp*_SARSr-F | GTGARATGGTCATGTGTGGCGG |
| | *RdRp*_SARSr-P2 | FAM-CAGGTGGAACCTCATCAGGAGATGC-BBQ |
| | *RdRp*_SARSr-R | CARATGTTAAASACACTATTAGCATA |
| *GAPDH* | *GAPDH*-F | GTGAAGGTCGGAGTCAACGG |
| | *GAPDH*-P | ROX-CGCCTGGTCAACAGGGTCGC-BBQ |
| | *GAPDH*-R | TCAATGAAGGGGTCATTGATG |

values lower than 37 and 40 in the green and orange channels, respectively, were reported as "positive".

All reactions were conducted using positive and negative controls, including those related to RNA extraction. The RNA from the original sample 28549 used to isolate and titer the Peruvian strain in the Vero cell line at the National Reference Laboratory of Respiratory Viruses was used as a positive control. The RNA from SARS-CoV-2 negative samples, whose results were obtained according to the standard protocol suggested by WHO, was used as a negative control. The PCR water was used as negative reaction control.

## Statistical analysis

The Ct values obtained from Real time RT-PCR reactions performed with both RNA extraction methods during the standardization proceeding and related to the comparison of six epidemiological weeks were submitted to Normality Test and submitted to ANOVA in both channels, *RdRp* and *GAPDH*. The p values < 0.05 and F values > F critic were considered significant differences.

## Results and discussion

The Real time RT-PCR reactions using both RNA extraction methods are shown in S1 Table, containing the sample code, Ct values for *RdRp* and *GAPDH* channels, average, and standard deviation. The standard deviation observed on the *GAPDH* channel was higher than the *RdRp* channel. The statistical analysis of variance was conducted from Ct values obtained in each channel. In the *RdRp* channel, the *p* and *F* values indicated that there was no significant difference between both RNA extraction methods, commercial kit versus in-house proteinase K, followed by thermal shock (*F = 0.653970542; p = 0.421036775; F critic = 3.957388322*; summary of these analyzes available in S1 Table). However, in the *GAPDH* channel, the p and F values presented a significant difference (*F = 15.27288168; p = 0.000139023; F critic = 3.902553068*; summary of these analyzes available in S1 Table). The F value represents how much the variability between the means can exceeds the expected value due the chance; in this sense, F values

**Table 2. Real time RT-PCR multiplex for SARS-CoV-2 using *GAPDH* as internal control: Conditions and thermal cycling reactions using biotechrabbit reagents.**

| Phases | Temperature | Time | Cycles |
|---|---|---|---|
| Reverse transcription | 50°C | 10 minutes | 1 |
| Initial denaturation | 95°C | 3 minutes | 1 |
| PCR amplification | 95°C | 10 seconds | 45 |
| | 58°C | 30 seconds | |
| | 40°C | 30 seconds | |

superior the F critical values is analogous to a p value less than alpha, indicating the rejection of the null hypothesis (that there is no significant difference between averages) [11].

Our biggest challenge after establishing the communitarian transmission of SARS-CoV-2 in Peru during the COVID-19 pandemic has been obtaining reagents and critical material to conduct and maintain the molecular testing. The privation of specific reagents' suppliers, for example, made it necessary to implement and validate protocols to guarantee the continuity of tests. Furthermore, establishing optimizations in the diagnostic tools for cost and processing time reduction is essential to make it possible to process sufficient samples in developing countries.

The COVID-19 molecular diagnosis can be biased by the samples' quality, the RNA extraction process, and factors related to the amplification protocol. To overcome these difficulties, the INS has conducted and validated several protocols from sample collection until final diagnosis (in house viral transport medium, saliva, diagnostic methods–molecular, serological, among others). In many countries, particularly in Latin America, the COVID-19 pandemic placed in evidence a flawed health system and a low response capacity for timely diagnosis that effectively contributes to surveillance and control. The WHO recommended molecular tests, but they were very scarce at the beginning, there were not enough laboratories, and the external market for diagnostic supplies was closed. In this context, there was a need to implement alternative techniques or adapt existing ones to make them more viable and reduce processing times and costs, and thus it was achieved by using thermal shock as part of the Real time RT-PCR processing for the diagnosis of SARS-CoV- 2 in Peru and can be extended to other countries with limited resources.

Technically, the RNA molecule, the genetic material of coronaviruses, is overly sensitive to the rise and drastic temperature variation used in our protocol. On the other hand, the results obtained by us and other researchers using protocols of kit-free RNA extraction have presented yielded promising results. Many of these have been made considering physic-chemistry characteristics commonly used in step from kit-based protocols. Kriegova et al. (2020) [12], for example, considered the temperature increase as physic tool and described the Direct-One-Step-RT-qPCR (DIOS-RT-qPCR) protocol for SARS-CoV-2 detection using 14 μL of the medium used to transport the swab previously submitted to virus inactivation at 75°C at 10 minutes; they obtained an efficient analytical sensibility, being able to detect virtually 550 virus copies/mL. Wozniak et al. (2020) [13], in contrast, also published a kit-free procedure for RNA extraction but based in chemistry principle, the pH change, presenting results scientifically acceptable.

Considering these shared physical-chemistry parameters associated in an unique protocol, Michel et al. (2020) [14] described the "COVID-quick-DET", in which the dry swab should be submitted to elution in 1 mL of normal saline, mixed using vortex during 5 seconds and treated with the same concentration of proteinase K used in our procedure. The main differences are related to an additional step, the centrifugation step (8,000 rpm, 2–5 seconds) and the absence of thermal shock (incubation in a dry thermal block: 56°C, 3 min; 95°C, 3 min). Other studies, many of them usually denominated "direct detection of SARS-CoV-2" via a RT-qPCR assay [15–17], have used steps pre-PCR similar to our protocol; they also have identified a high percentage of agreement in paired analysis in that the tested methods were compared with standard commercial kit protocols. In general, the dissonant dataset of samples has been associated with low viral load (high Ct), and which was not observed in our study. The pre-PCR protocol of Smirlak et al. [18] is based on heat-inactivated or lysed samples; they displayed the effectiveness of heat inactivation by plaque assay and the use of alternative buffers as a transport medium for direct Real time RT-PCR; their results were significant and comparable to ours by saving time and cost in molecular testing for COVID-19.

**Table 3. Comparison of COVID-19 positivity obtained by Real time RT-PCR multiplex for SARS-CoV-2 using *GAPDH* as internal control from two RNA extraction methods in six epidemiological weeks, four using commercial kits (EW 21, 22, 28 and 29) and two using proteinase k treatment followed by thermal shock (EW 24 and 25).**

| Epidemiological week | Samples processed | Positive (n) | Positivity (%) | Average (%) | SD |
|:---:|:---:|:---:|:---:|:---:|:---:|
| 21 | 17,212 | 5,275 | 30.65 | 31.59 | 1.329361 |
| 22 | 20,287 | 6,599 | 32.53 | | |
| 24 | 26,160 | 8,042 | 30.74 | 30.955 | 0.304056 |
| 25 | 23,238 | 7,243 | 31.17 | | |
| 28 | 25,134 | 8,343 | 33.19 | 32.8 | 0.551543 |
| 29 | 36,365 | 11,787 | 32.41 | | |
| Total | 148,396 | 47,289 | 31.78 | | |

We performed comparative experiments in triplicate, in the laboratory standardization, and performed the statistical analysis using the averages, finding 100% agreement in our results. In addition, our statistical analyses showed no significant difference between Ct values obtained when the same sample was processed in parallel by both RNA extraction methods in the green channel, e.g., for *RdRp* detection (SARS-CoV-2 specific). Nevertheless, there was significant difference between RNA extracted using our method compared to the one that used the commercial kit in the orange channel, e.g., *GAPDH* channel (internal control), in which the variance of Ct values was higher in samples submitted to extraction using commercial kit than that extracted using our method. Common RNA purification kits have, in contrast, a step for the removal of genomic DNA. Our non-kit-purified protocol does not contain any DNase step, resulting in higher amounts of cellular debris and contaminating DNA in the solution. The extra material is not a problem for the viral targets but can explain significant variations in *GAPDH* detection.

Last and not least, we performed a comparison of positivity between epidemiological weeks (EW) in June-2020. In that moment, the INS of Peru processed 148,396 clinical samples during the EW 21, 22, 24, 25, 28, and 29. In this period, the samples had their RNA extracted using commercial kits on 21, 22, 28, and 29 EW and using exclusively proteinase K treatment followed by thermal shock on 24 and 25 epidemiological weeks. The total of samples processed, and the positivity presented normal distribution. The comparison was established from the positivity average between these weeks, and there was no significant difference, and the data were showed in Table 3.

The more significant number of samples processed in the last two (EW 28 and 29) and the increase in positivity may be associated with the quarantine ending on 30th June. However, this change in behavior in the population did not lead to a significant increase in positivity. Even that several protocols have been published about molecular testing for COVID-19, and all of them have similar principles and methods, involving heating for viral inactivation, buffer solutions for samples transporting and optimizations of PCR protocols, we did not find any research describing a screening in the population. Our findings can be unconclusive but similarly could suggest that our protocol kit-free presented analogous yield with the reference protocol kit-based.

All these findings have indicated that alternative methods present adequate sensitivity to detect positive patients potentially spreaders. Besides, the processing time was faster using our protocol in which the time of extraction was optimized to be done in almost 30 minutes, including hand pipetting. This point also could be an imperative feature because consequently more people were tested simultaneously. Additionally, these proceeding of RNA extraction

could be tested using other enveloped viruses and could be a feasible option to be implemented in the diagnostic routine in public-health surveillance of infectious diseases.

## Conclusions

We concluded that our in-house method is an easy, fast, and cost-effective alternative method for extracting RNA and conducing molecular diagnosis of COVID-19.

## Supporting information

**S1 Appendix. Real time RT-PCR multiplex for SARS-CoV-2 using GAPDH as internal control.**
(DOCX)

**S1 File.**
(DOCX)

**S1 Table. Comparative COVID-19 molecular diagnosis by Real time RT-PCR multiplex for SARS-CoV-2 using *GAPDH* as internal control from two RNA extraction methods.** The comparison was established from Ct values of two channels, *RdRp* specific for SARS-CoV-2 and *GAPDH*, obtained in each sample extracted by both RNA extraction methods.
(XLSX)

## Acknowledgments

We are thankful to all the Microbiology and Biomedicine Laboratory workers of INS, who were involved in obtaining, handling, and processing the samples for COVID-19 molecular diagnosis.

## Author Contributions

**Conceptualization:** Adolfo Marcelo Ñique, Jairo Andrés Mendez Rico, Cesar Cabezas Sanchez, Jan Felix Drexler.

**Data curation:** Jairo Andrés Mendez Rico, Paulo Vitor Marques Simas.

**Formal analysis:** Paulo Vitor Marques Simas, Jan Felix Drexler.

**Investigation:** Adolfo Marcelo Ñique, Fiorella Coronado-Marquina, María Paquita García Mendoza, Jan Felix Drexler.

**Methodology:** Adolfo Marcelo Ñique, Jairo Andrés Mendez Rico, María Paquita García Mendoza, Jan Felix Drexler.

**Project administration:** Jairo Andrés Mendez Rico, Cesar Cabezas Sanchez.

**Resources:** Jairo Andrés Mendez Rico, Cesar Cabezas Sanchez.

**Supervision:** Jairo Andrés Mendez Rico, Nancy Rojas-Serrano, Cesar Cabezas Sanchez, Jan Felix Drexler.

**Validation:** Adolfo Marcelo Ñique, Fiorella Coronado-Marquina, Nancy Rojas-Serrano.

**Visualization:** Nancy Rojas-Serrano.

**Writing – original draft:** Paulo Vitor Marques Simas, Jan Felix Drexler.

**Writing – review & editing:** Jairo Andrés Mendez Rico, Paulo Vitor Marques Simas, Jan Felix Drexler.

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
