## [Decision Letter · Decision Letter 0]

3 Nov 2020

PONE-D-20-28949

Proteinase K treatment followed by thermal shock as faster, feasible and low cost RNA extraction alternative for molecular detection of SARS-CoV-2

PLOS ONE

Dear Dr. Paulo V Marques Simas, 

Thank you for submitting your manuscript to PLOS ONE. After careful consideration, we feel that it has merit but does not fully meet PLOS ONE’s publication criteria as it currently stands. Therefore, we invite you to submit a revised version of the manuscript that addresses the points raised during the review process.

We look forward to receiving your revised manuscript.

Kind regards,

Shawky M. Aboelhadid, PhD

Academic Editor

PLOS ONE

Journal Requirements:

2. Please include your tables as part of your main manuscript and remove the individual files. Please note that supplementary tables should be uploaded as separate "supporting information" files.

3. We noted in your submission details that a portion of your manuscript may have been presented or published elsewhere.

"The article contains informations related to "RT-qPCR Multiplex using GAPDH as internal control to detect SARS-CoV-2" submitted to Infectious Diseases and Therapy; this paper is being evaluated by Editor-in-chief."

Reviewers' comments:

Reviewer's Responses to Questions

**Comments to the Author**

1. Is the manuscript technically sound, and do the data support the conclusions?

Reviewer #1: Partly

Reviewer #2: Yes

Reviewer #3: Partly

2. Has the statistical analysis been performed appropriately and rigorously? 

Reviewer #1: Yes

Reviewer #2: Yes

Reviewer #3: I Don't Know

3. Have the authors made all data underlying the findings in their manuscript fully available?

Reviewer #1: Yes

Reviewer #2: No

Reviewer #3: Yes

4. Is the manuscript presented in an intelligible fashion and written in standard English?

Reviewer #1: Yes

Reviewer #2: Yes

Reviewer #3: No

5. Review Comments to the Author

Reviewer #1: In this study, the authors compared the performance of proteinase K treatment followed by thermal shock with commercial extraction methods, which provides an alternative way to test SARS-CoV-2 in low income regions. The comments discussed below are provided for consideration.

1. The authors used three different enzymes and reagents to perform PCR, but didn’t mention the differences between these reagents. Could the discrepancy between results be due to the reagents used?

2. Table 3: How many times was each sample repeated? Where does average and SD come from?

3. As seen in Table 3, although the statistical analysis of variance showed no significant difference, the CT value of several samples is largely different in commercial kits group and PK+TS group. Such as 84667, 84677, 85480, 85481…

4. As for testing RNA virus, RNaseP might be better than GAPDH as internal control.

5. Line 283, is it possible that proteinase K treatment caused false positive result? The inconsistent results should be confirmed with a more sensitive reagent. Also, the successful amplification of GAPDH cannot guarantee that there were no false negative results.

6. Table 4, the positive rate of week 24 and 25 is slightly lower than other weeks.

Reviewer #2: This article is timely in the current pandemic, with many readers likely keen to learn as much about COVID-19 as possible. However, there are some major issues with this article.

Title: suitable

Abstract: The abstract should contain a mention of the specific study design and at the end should contain a sentence on implication of this study.

Background: The background is adequate

Methods: The methods section should be revisited. Is this indeed a retrospective study? In a retrospective study of how the two methods were compared. How was the sample compared before and after archiving? What is the commercial kit used in this study?

Results: The results are adequate for a retrospective study

Discussion: There is need to discuss these findings to reflect the objectives of the study. Authors should focus on these and compare their results critically with other similar studies and derive a valid inference based on their findings.

Conclusion: adequate for a retrospective study

References: inadequate

General comments: The whole manuscript needs copy-editing and should preferably be edited by an English expert or native English speaker

Reviewer #3: The presented manuscript by Ñique and colleagues presents a helpful addition to the growing list of publications describing alternatives to kit-based RNA isolation for SARS-CoV-2 diagnostics, which is of continuing importance considering increasing case numbers in secondary waves of infection in countries with limited resources. Their side-by-side comparison of kit-based and kit-free RNA preparation with SARS-CoV-2 and endogenous control target detection is sound and shows that ProtK + Heat shock is a suitable alternative to RNA purification if resources are limited. However, the article suffers from poor English in some places, making it less comprehensible. The presented method should also be compared to already published kit-free protocols for SARS-CoV-2 qPCR rather than unrelated protocols for DNA processing. Some conclusions need to be revised – there cannot be 100% concordance between the protocols if at least 1/72 samples is positive in one protocol but not the other, as discussed. Comparing epidemiological weeks when kit-based and kit-free detection was performed based on positivity rates is interesting, but it should be added that this cannot be proof of whether the method is superior/equivalent or not, as there is no direct comparison for the same samples. A revised version would be suitable for publication in PLOS ONE.

Typo in financial disclosure “associated whit Ministry of Health”

Language - in competing interests “declare none” – should be “no”

Line 1: Title “faster” – than what? Change to fast or include “kit-based” or something like this

Line 26: Change sentence structure “Sometimes, even with the funds and resources available to purchase them, the lack of suppliers and the global shortage of material as consequence of the high demand, makes timely detection and diagnosis of the virus carrier difficult.”

Line 34: Concordance was 100% and then RT-qPCR was performed” – concordance with test samples? Add this, otherwise confusing.

Line 37: This sentence is confusing and only made sense after reading the entire manuscript, so it should be formulated more clearly. I would suggest the following: “We compared the SARS-CoV-2 positivity obtained using our kit-free method within two complete epidemiological weeks to the two weeks prior and after, where RNA was extracted using commercial kits. There was no significant difference between positivity rates, suggesting the use of a kit-free RNA preparation method did not significantly alter detection rates.”

Line 41: Language – “there was no significant difference”. Also, what does “positive results average” mean?

Line 49: “type” is not part of the name of the virus, it’s just “Severe acute respiratory syndrome coronavirus 2”

Line 54: Language – change to “and makes them”

Line 65: 5-log reduction in what – infectivity? Please add.

Line 66: Language – change to “are also used during amplification by RT-qPCR”

Line 67: Language – change to “Furthermore, using an internal control for RT-qPCR reaction can confirm integrity of the samples, decreasing the probability of false-negative results” (“exponentially” is odd here)

Line 70: There’s something missing in this sentence. I assume it should be something like “… physical methods like heating and specific chemical reagents like proteinase K widely used in genetic material extraction protocols may be used to optimize viral purification”

Line 76: Remove “assay” – a purification is not an assay.

Line 77: Spelling “stablished” – change to “established”

Line 79: Language – change to “We compared to efficiency of the new method based on Ct values obtained by…”

Line 91: Language – I assumed “people suspected” should just be “patients”

Line 92: Language – change to “sample collection”

Line 132-147: This paragraph is confusing. You tested 72 samples with both methods. Also, during June 3-25, all diagnostics was done with the presented ProtK/thermal shock method because no kits were available. Was this before or after the 72 samples were tested with both methods? Then you’re comparing positivity obtained in the period where kit-less methods were used to other periods where kit-based methods were used – on the assumption that numbers would be similar based on progression (or rather non-progression) of the pandemic? This is interesting, but should not be over-interpreted and treated with caution.

Line 181: Language – “in both channels”

Line 198: Here you write that ANOVA was used on Ct values, while in the methods section (line 181) you write that “qualitative results of RT-qPCR were submitted to ANOVA”. Ct is a quantitative result.

Line 202: You should discuss why GAPDH is significantly more abundant in the non-kit samples. The differences is quite substantial – I would suspect this is from cellular debris that may be more abundant in the non-kit-purified samples.

Line 223: Language/confusing – there was an increase in cases (and testing), likely due to the lockdown being lifted but the increase was not significant compared to previous weeks? If so, rephrase to “this change in behaviour in the population apparently didn’t lead to a significant increase in positivity”.

Line 226: Spelling “stablishing” – change to establishing.

Line 229: Language – change “provided” to “made it necessary to implement and validate protocols in house…”

Line 232: Language – change to “is essential to make it possible to process sufficient samples in underdeveloped countries”

Line 234: Language – “sample quality”

Line 239: It’s okay to mention this but DNA from spores and RNA from viruses are hardly comparable, so I would remove some of the details that don’t add anyhting to this manuscript.

Line 254: Language – “since the samples are processed in 96 well plate format”. Also, especially since this is a crucial point, just state the time/sample. Currently it sounds odd that 5.5h is “8 times higher” than “less than 1h”.

Line 257: Similar to the spores, not really relevant to compare environmental DNA to viral RNA.

Line 264: It is also okay to cite preprints. Besides there are multiple published manuscripts describing “kit-free” RNA preparation for qPCR of SARS-CoV-2 that should be discussed.

10.1016/j.jviromet.2020.113965

10.3390/diagnostics10080605

https://doi.org/10.1038/s41598-020-73616-w

https://doi.org/10.1371/journal.pbio.3000896

https://doi.org/10.1371/journal.pone.0236564

https://doi.org/10.1038/s41467-020-18611-5

https://doi.org/10.1016/j.jcv.2020.104423

Line 274: Language – “since these variables”

Line 276: Why the quotation marks? Also its rather what makes this “purification” possible, not an advantage.

Line 283: Earlier in the manscript you write about “100% concordance”, yet here is a sample where this is not the case. I would also caution the interpretation that a sample negative by definition of Cut-off when purified with the commercial kit but positive when treated with ProtK/thermal shock was “false-negative” with the commercial kits – both values are very high and I would be cautious about higher background signal in non-purified samples.

Line 291: But the advantage of multiplexed detection of SARS-CoV-2 and an endogenous control would be the case for both the commercial kit and your in-house method. And you cannot exclude false-positives by your method compared to commercial kits based on higher background.

Line 292: ”Our results showed no significant difference between Ct values obtained when the same sample were processed in parallel by both RNA extraction” – what about sample 84677 mentioned just in the above paragraph?

Line 294: “Ct values of positive control have always been in agreement” – but this is already isolated RNA. Did you also subject the sample sample to your in-house method of ProtK/Heat?

Line 301: Would be highly cautious calling something a “better yield” based on one sample where this happened (see above comment)

Table 2: Add the actual product used somewhere (catalogue ID).

Table 3: It would be helpful if this was also displayed graphically. Also please state what the averages are derived from – commercial kit and PK + TS?

6. PLOS authors have the option to publish the peer review history of their article (what does this mean?). If published, this will include your full peer review and any attached files.

Reviewer #1: No

Reviewer #2: No

Reviewer #3: No

---

## [Author Response · Author response to Decision Letter 0]

14 Dec 2020

REPONSE TO THE EDITOR AND REVIEWERS

PONE-D-20-28949

#EDITOR:

R: The data related to the standardization of RT-qPCR cited in our paper is from the INS and only was used as a useful tool to test our RNA extraction method. Since these data are from INS, this does not constitute a dual publication. 

 

Reviewer #1: In this study, the authors compared the performance of proteinase K treatment followed by thermal shock with commercial extraction methods, which provides an alternative way to test SARS-CoV-2 in low income regions. The comments discussed below are provided for consideration.

1. The authors used three different enzymes and reagents to perform PCR, but didn’t mention the differences between these reagents. Could the discrepancy between results be due to the reagents used?

R: The enzymes and reagents mentioned were used in the preliminary tests, considering the standard procedures for implementing a new INS protocol (data not relevant). The Biotech Rabbit was used in the RT-qPCR reactions to compare only the extraction methods. Since we conducted new RT-qPCR reactions for replicates, we only used a type of enzyme, the Biotech Rabbit, to standardize the reactions and consider only the comparison between the RNA extraction methods.

2. Table 3: How many times was each sample repeated? Where does average and SD come from?

R: We conducted new RT-qPCR reactions in triplicate. The new data were updated and submitted as table S1.

3. As seen in Table 3, although the statistical analysis of variance showed no significant difference, the CT value of several samples is largely different in commercial kits group and PK+TS group. Such as 84667, 84677, 85480, 85481…

R: The new statistical analysis showed a significant difference between RNA extracted using our method compared to that used commercial kit. Also, we identified higher variance between samples when extracted by commercial kit than that extracted using our method. This point could be justified by the column method that uses solid phase and can saturate. Since the internal control is more abundant than viral RNA and our method considers total RNA in the solution, the saturation does not occur, and the RT-qPCR reactions can detect the GAPDH RNA more easily (this material is widely available).

4. As for testing RNA virus, RNaseP might be better than GAPDH as internal control.

R: We agree with this point. The GAPDH gene was already used as an internal control for the detection of SARS COV2. Besides, the supplies were available, and their cost is lower compared to commercial RT-PCR kits.

5. Line 283, is it possible that proteinase K treatment caused false positive result? The inconsistent results should be confirmed with a more sensitive reagent. Also, the successful amplification of GAPDH cannot guarantee that there were no false negative results.

R: We performed the RNA extraction from the same sample using both methods, and the results were 100% in agreement. In this sense, the samples with low viral load should present disagreement results between the methods (the commercial kit should be more sensible), which was not valid. We understood that our protocol presented the same efficiency as that commercial kit to extract the SARS-CoV-2 RNA.

6. Table 4, the positive rate of week 24 and 25 is slightly lower than other weeks.

R: Even the positivity data presented in the epidemiological weeks 24 and 25 was slightly lower than the others. There was none significant difference between them.

 

Reviewer #2: This article is timely in the current pandemic, with many readers likely keen to learn as much about COVID-19 as possible. However, there are some major issues with this article.

Title: suitable 

Abstract: The abstract should contain a mention of the specific study design and at the end should contain a sentence on implication of this study.

Background: The background is adequate

Methods: The methods section should be revisited. Is this indeed a retrospective study? In a retrospective study of how the two methods were compared. How was the sample compared before and after archiving? What is the commercial kit used in this study?

Results: The results are adequate for a retrospective study

Discussion: There is need to discuss these findings to reflect the objectives of the study. Authors should focus on these and compare their results critically with other similar studies and derive a valid inference based on their findings.

Conclusion: adequate for a retrospective study

References: inadequate

General comments: The whole manuscript needs copy-editing and should preferably be edited by an English expert or native English speaker

 

Reviewer #3: The presented manuscript by Ñique and colleagues presents a helpful addition to the growing list of publications describing alternatives to kit-based RNA isolation for SARS-CoV-2 diagnostics, which is of continuing importance considering increasing case numbers in secondary waves of infection in countries with limited resources. Their side-by-side comparison of kit-based and kit-free RNA preparation with SARS-CoV-2 and endogenous control target detection is sound and shows that ProtK + Heat shock is a suitable alternative to RNA purification if resources are limited. However, the article suffers from poor English in some places, making it less comprehensible. The presented method should also be compared to already published kit-free protocols for SARS-CoV-2 qPCR rather than unrelated protocols for DNA processing. Some conclusions need to be revised – there cannot be 100% concordance between the protocols if at least 1/72 samples are positive in one protocol but not the other, as discussed. Comparing epidemiological weeks when kit-based and kit-free detection was performed based on positivity rates is interesting, but it should be added that this cannot be proof of whether the method is superior/equivalent or not, as there is no direct comparison for the same samples. A revised version would be suitable for publication in PLOS ONE.

Typo in financial disclosure “associated whit Ministry of Health”

Language - in competing interests “declare none” – should be “no”

Line 1: Title “faster” – than what? Change to fast or include “kit-based” or something like this

R: The title was updated according to your suggestion.

Line 26: Change sentence structure “Sometimes, even with the funds and resources available to purchase them, the lack of suppliers and the global shortage of material as consequence of the high demand, makes timely detection and diagnosis of the virus carrier difficult.”

R: Sometimes, the lack of suppliers and the global shortage of material due to the high demand make it difficult for timely detection and diagnosis of the virus carrier even with the funds and resources available to purchase the reagents.

Line 34: Concordance was 100% and then RT-qPCR was performed” – concordance with test samples? Add this, otherwise confusing.

R: The concordance between results obtained by RT-qPCR from both RNA extraction methods was 100% in the samples tested. In the data presented in table 3, there was a disagreement related to only one sample. Since one of the reviewers suggested that we present results obtained from average, we executed new rounds of experiments, considering experimental triplicate and including other six samples more (from 72 samples to 78 samples; from unique running to triplicate running). The new results presented a 100% in agreement.

Line 37: This sentence is confusing and only made sense after reading the entire manuscript, so it should be formulated more clearly. I would suggest the following: “We compared the SARS-CoV-2 positivity obtained using our kit-free method within two complete epidemiological weeks to the two weeks prior and after, where RNA was extracted using commercial kits. There was no significant difference between positivity rates, suggesting the use of a kit-free RNA preparation method did not significantly alter detection rates.”

We make all changes according to the suggestions.

Line 41: Language – “there was no significant difference”. Also, what does “positive results’ average” mean?

Line 49: “type” is not part of the name of the virus, it’s just “Severe acute respiratory syndrome coronavirus 2”

Line 54: Language – change to “and makes them”

Line 65: 5-log reduction in what – infectivity? Please add.

R: The information was added.

Line 66: Language – change to “are also used during amplification by RT-qPCR”

Line 67: Language – change to “Furthermore, using an internal control for RT-qPCR reaction can confirm integrity of the samples, decreasing the probability of false-negative results” (“exponentially” is odd here)

Line 70: There’s something missing in this sentence. I assume it should be something like “… physical methods like heating and specific chemical reagents like proteinase K widely used in genetic material extraction protocols may be used to optimize viral purification”

Line 76: Remove “assay” – a purification is not an assay.

R: The word was removed.

Line 77: Spelling “stablished” – change to “established”

Line 79: Language – change to “We compared to efficiency of the new method based on Ct values obtained by…”

Line 91: Language – I assumed “people suspected” should just be “patients”

Line 92: Language – change to “sample collection”

Line 132-147: This paragraph is confusing. You tested 72 samples with both methods. Also, during June 3-25, all diagnostics was done with the presented ProtK/thermal shock method because no kits were available. 

1. Was this before or after the 72 samples were tested with both methods? 

R: These proceedings for COVID-19 molecular diagnosis using RNA extracted using heat shock and proteinase k were conducted after validation using the 78 samples.

2. Then you’re comparing positivity obtained in the period where kit-less methods were used to other periods where kit-based methods were used – on the assumption that numbers would be similar based on progression (or rather non-progression) of the pandemic? This is interesting, but should not be over-interpreted and treated with caution.

R: The sentence was rewritten considering this caution.

Line 181: Language – “in both channels”

Line 198: Here you write that ANOVA was used on Ct values, while in the methods section (line 181) you write that “qualitative results of RT-qPCR were submitted to ANOVA”. Ct is a quantitative result. 

R: We conducted experiments without using a standard curve. In our case, the Ct values were only indirect measures of viral load.

Line 202: You should discuss why GAPDH is significantly more abundant in the non-kit samples. The differences are quite substantial – I would suspect this is from cellular debris that may be more abundant in the non-kit-purified samples.

Line 223: Language/confusing – there was an increase in cases (and testing), likely due to the lockdown being lifted but the increase was not significant compared to previous weeks? If so, rephrase to “this change in behavior in the population apparently didn’t lead to a significant increase in positivity”.

Line 226: Spelling “stablishing” – change to establishing.

Line 229: Language – change “provided” to “made it necessary to implement and validate protocols in house…”

Line 232: Language – change to “is essential to make it possible to process sufficient samples in underdeveloped countries”

Line 234: Language – “sample quality”

Line 239: It’s okay to mention this but DNA from spores and RNA from viruses are hardly comparable, so I would remove some of the details that don’t add anything to this manuscript.

Line 254: Language – “since the samples are processed in 96 well plate format”. Also, especially since this is a crucial point, just state the time/sample. Currently it sounds odd that 5.5h is “8 times higher” than “less than 1h”.

Line 257: Similar to the spores, not really relevant to compare environmental DNA to viral RNA.

Line 264: It is also okay to cite preprints. Besides there are multiple published manuscripts describing “kit-free” RNA preparation for qPCR of SARS-CoV-2 that should be discussed.

These papers should be included in the discussion.

https://doi.org/10.1016/j.jviromet.2020.113965

https://doi.org/10.3390/diagnostics10080605

https://doi.org/10.1038/s41598-020-73616-w

https://doi.org/10.1371/journal.pbio.3000896

https://doi.org/10.1371/journal.pone.0236564

https://doi.org/10.1038/s41467-020-18611-5

https://doi.org/10.1016/j.jcv.2020.104423

R: All these papers suggested were included in the discussion section.

Line 274: Language – “since these variables”

Line 276: Why the quotation marks? Also its rather what makes this “purification” possible, not an advantage. 

R: The quotation marks was removed.

Line 283: Earlier in the manuscript you write about “100% concordance”, yet here is a sample where this is not the case. I would also caution the interpretation that a sample negative by definition of Cut-off when purified with the commercial kit but positive when treated with ProtK/thermal shock was “false-negative” with the commercial kits – both values are very high and I would be cautious about higher background signal in non-purified samples.

R: Here we can explain that the RT-qPCR protocol used in our study was submitted to the validation process, and the positivity was measured from this point, e.g., from the cut-off obtained in that validation. There was no relation to the results presented in our paper about heat shock and proteinase k for RNA extraction.

Line 291: But the advantage of multiplexed detection of SARS-CoV-2 and an endogenous control would be the case for both the commercial kit and your in-house method. And you cannot exclude false-positives by your method compared to commercial kits based on higher background.

Line 292: “Our results showed no significant difference between Ct values obtained when the same sample were processed in parallel by both RNA extraction” – what about sample 84677 mentioned just in the above paragraph?

Line 294: “Ct values of positive control have always been in agreement” – but this is already isolated RNA. Did you also subject the sample to your in-house method of ProtK/Heat?

R: In this case, we think this point is not necessary because positive control is a standard. Many researchers use g-block or other synthetic genetic material as a positive control in their RT-qPCR reactions. Furthermore, the experiments were conducted using clinical samples, which should be considered more critical.

Line 301: Would be highly cautious calling something a “better yield” based on one sample where this happened (see above comment)

R: The expression was removed.

Table 2: Add the actual product used somewhere (catalogue ID).

R: This information was updated.

Table 3: It would be helpful if this was also displayed graphically. Also please state what the averages are derived from – commercial kit and PK + TS?

R: This table was removed and submitted as support material (table S1).

---

## [Decision Letter · Decision Letter 1]

24 Dec 2020

PONE-D-20-28949R1

Proteinase k treatment followed by thermal shock as faster, more feasible, and lower cost than RNA extraction kit-based as an alternative for molecular detection of SARS-CoV-2

PLOS ONE

Dear Dr. Paulo Vitor Marques Simas,

Thank you for submitting your manuscript to PLOS ONE. After careful consideration, we feel that it has merit but does not fully meet PLOS ONE’s publication criteria as it currently stands. Therefore, we invite you to submit a revised version of the manuscript that addresses the points raised during the review process.

ACADEMIC EDITOR: Please  authors, you must be replied to the reviewer comments. The manuscript should be revised by expertise in English language. 

We look forward to receiving your revised manuscript.

Kind regards,

Shawky M. Aboelhadid, PhD

Academic Editor

PLOS ONE

Reviewers' comments:

Reviewer's Responses to Questions

**Comments to the Author**

1. If the authors have adequately addressed your comments raised in a previous round of review and you feel that this manuscript is now acceptable for publication, you may indicate that here to bypass the “Comments to the Author” section, enter your conflict of interest statement in the “Confidential to Editor” section, and submit your "Accept" recommendation.

Reviewer #2: (No Response)

Reviewer #3: (No Response)

2. Is the manuscript technically sound, and do the data support the conclusions?

Reviewer #2: Partly

Reviewer #3: Yes

3. Has the statistical analysis been performed appropriately and rigorously? 

Reviewer #2: Yes

Reviewer #3: I Don't Know

4. Have the authors made all data underlying the findings in their manuscript fully available?

Reviewer #2: Yes

Reviewer #3: Yes

5. Is the manuscript presented in an intelligible fashion and written in standard English?

Reviewer #2: No

Reviewer #3: No

6. Review Comments to the Author

Reviewer #2: Dear Editor

Unfortunately, the author did not answer the questions properly

Reviewer #3: The revised manuscript is improved in several areas, but there are still many language problems. The manuscript needs proof-reading by someone proficient in English. I will only point out the major language problems specifically here. Scientifically, the discussion is greatly improved but would benefit from more careful examination of the cited literature and less exaggerated conclusions regarding superiority of the presented method.

The title already has odd sentence structure.

Change: “Proteinase k treatment followed by thermal shock as faster, more feasible, and lower cost than RNA extraction kit-based as an alternative for molecular detection of SARS- CoV-2“ to “Proteinase K treatment followed by thermal shock as a faster, more feasible and lower-cost alternative to RNA extraction kit-based purification for molecular detection of SARS-CoV-2.

Line 54: “the variance was higher in the samples extracted by commercial kit than that used the alternative protocol,suggesting indirectly that our protocol was more efficient to total RNA extraction” – this is the least likely explanation for this observation. Much more likely, this is from cellular debris present in the non-kit samples (possibly even DNA).

Line 306: This is incorrect, Michel et al. did use clinical samples; SARS-CoV-2 virus stocks were only applied to differentiate sensitivity of the methods.

Line 313: “was really good” is a not very scientific term

Line 408: “This point could be justified by the column method uses solid phase and can saturate” – I think I know what you mean but this sentence is poorly phrased. I would also doubt the binding capacity of RNA purification columns is a realistic problem for swab material. Typical RNA mini columns (Qiagen) can bind 100 μg RNA. This is equivalent to millions of cells, which are certainly not present in swab material. The more likely explanation is that the non-kit-purified samples contain higher amounts of cellular debris and contaminating DNA. Unless I’m missing it, your protocol does not contain any DNase step that would get rid of cellular DNA contamination. Obviously this is not a problem for the viral targets, but can explain variations in GAPDH detection. Common RNA purification kits have, in contrast, a step for removal of genomic DNA.

Line 445: “Our protocol, additionally to solve our temporary” – is something missing here?

Line 448: “More than a new method, our protocol seems to be a large vantage as a public health approach” - I assume you meant to write advantage, but that’s a pretty strong statement regardless, especially with many similar protocols published.

Response to rebuttal:

Line 34: Concordance was 100% and then RT-qPCR was performed” – concordance with test samples? Add this, otherwise confusing.

R: The concordance between results obtained by RT-qPCR from both RNA extraction methods was 100% in the samples tested. In the data presented in table 3, there was a disagreement related to only one sample. Since one of the reviewers suggested that we present results obtained from average, we executed new rounds of experiments, considering experimental triplicate and including other six samples more (from 72 samples to 78 samples; from unique running to triplicate running). The new results presented a 100% in agreement

Reviewer response: Are these new samples? Why is the numbering different than in the initial manuscript? I was curious what happened to sample 84677, which was positive in the PK+TS method and negative in the commercial kit method in the original manuscript. I would suggest showing the comparison of Ct obtained by both kits as a dot plot with regression line, so one can immediately see how similar it looks (for both the viral and control target).

2. Then you’re comparing positivity obtained in the period where kit-less methods were used to other periods where kit-based methods were used – on the assumption that numbers would be similar based on progression (or rather non-progression) of the pandemic? This is interesting, but should not be over-interpreted and treated with caution.

R: The sentence was rewritten considering this caution.

Reviewer response: I don’t see any changes reducing the conclusions on this. The manuscript submitted is also not fully change-tracked, as there are changes here but none changing the conclusions.

Line 198: Here you write that ANOVA was used on Ct values, while in the methods section (line 181) you write that “qualitative results of RT-qPCR were submitted to ANOVA”. Ct is a quantitative result.

R: We conducted experiments without using a standard curve. In our case, the Ct values were only indirect measures of viral load.

Reviewer response: This is fine, but Ct is still a quantitative measurement. You’re not comparing “positive vs. negative” in the ANOVA, so it’s still a comparison of quantiative data.

Line 301: Would be highly cautious calling something a “better yield” based on one sample where this happened (see above comment) R: The expression was removed.

Reviewer response: Not really, you’re still making the same point in Line 408 as mentioned above. I will believe this if you show it’s not DNA contamination; but you can also just tone down this conclusion.

7. PLOS authors have the option to publish the peer review history of their article (what does this mean?). If published, this will include your full peer review and any attached files.

Reviewer #2: No

Reviewer #3: No

---

## [Author Response · Author response to Decision Letter 1]

5 Feb 2021

Response to Reviewers

Please authors, you must be replied to the reviewer comments. The manuscript should be revised by expertise in English language. 

R: The authors carefully replied to all questions from the reviewers. In addition, the protocol is now available in https://doi.org/10.17504/protocols.io.br6bm9an. 

 

Response to the Reviewer #3:

1. The revised manuscript is improved in several areas, but there are still many language problems. The manuscript needs proof-reading by someone proficient in English. I will only point out the major language problems specifically here. Scientifically, the discussion is greatly improved but would benefit from more careful examination of the cited literature and less exaggerated conclusions regarding superiority of the presented method.

R: The paper was submitted to a careful review by a native English professor, and the discussion and the conclusions were restructured accordingly, suggested by the reviewer.

2. The title already has odd sentence structure.

R: The paper title was changed to “A faster and less costly alternative for RNA extraction of SARS-CoV-2 using Proteinase k treatment followed by thermal shock”.

3. Abstract: 

“the variance was higher in the samples extracted by commercial kit than that used the alternative protocol, suggesting indirectly that our protocol was more efficient to total RNA extraction”.

R: This sentence was deleted.

4. The misinformation about the experimental design of Michel and collaborators (2020) was eliminated. 

5. The term “was really good” was deleted.

6. The sentence “Common RNA purification kits have, in contrast, a step for the removal of genomic DNA. Our non-kit-purified protocol does not contain any DNase step, resulting in higher amounts of cellular debris and contaminating DNA in the solution. The extra material is not a problem for the viral targets but can explain significant variations in GAPDH detection", according to the reviewer's suggestion.

7. The sentence “Our protocol, additionally to solve our temporary” was deleted.

8. The sentence “More than a new method, our protocol seems to be a large vantage as a public health approach” was deleted. 

 

Response to the Reviewer #2:

1. Questions and answers related to the new round of experiments.

a) Are these new samples? 

R: The experiments were repeated under the same conditions, using the same range of Ct values and in experimental triplicate, but using a new round of samples. This change occurred because either the original samples were no longer available in our sample-bank or the RNA was degraded, preventing the new experiments, according to the reviewer's suggestion. 

b) Why is the numbering different than in the initial manuscript? 

R: Seeing we had to conduct new experiments for guaranteeing our protocol yield and considering the discordant result presented in sample 84677, we included six more samples (almost 10% increase). Unfortunately, sample 84677 was no longer available and therefore not included in the experimental replicates. I would suggest showing the comparison of Ct obtained by both kits as a dot plot with regression line, so one can immediately see how similar it looks (for both the viral and control target).

R: The dot plot with the regression line related to the comparison of the Ct values (viral target – RdRp gene: RNA extracted by our protocol versus RNA extracted using commercial kit; internal control – GAPDH gene: extracted by our protocol versus RNA extracted using the commercial kit) was produced and is now available bellow.

 

2. Then you’re comparing positivity obtained in the period where kit-less methods were used to other periods where kit-based methods were used – on the assumption that numbers would be similar based on progression (or rather non-progression) of the pandemic? This is interesting but should not be over-interpreted and treated with caution.

R: The sentence was rewritten considering this caution.

3. I don’t see any changes reducing the conclusions on this. The manuscript submitted is also not fully change-tracked, as there are changes here but none changing the conclusions.

R: The conclusion was rewritten according to both reviewers' suggestions, and the file was carefully change-tracked.

4. This is fine, but Ct is still a quantitative measurement. You’re not comparing “positive vs. negative” in the ANOVA, so it’s still a comparison of quantitative data.

R: The word “qualitatively” was deleted.

5. Would be highly cautious calling something a “better yield” based on one sample where this happened (see above comment) R: The expression was removed. Reviewer response: Not really, you’re still making the same point in Line 408 as mentioned above. I will believe this if you show it’s not DNA contamination; but you can also just tone down this conclusion.

R: The idea was restructured in all the papers, and these cautious were carefully considered by the authors.

---

## [Decision Letter · Decision Letter 2]

17 Feb 2021

PONE-D-20-28949R2

A faster and less costly alternative for RNA extraction of SARS-CoV-2 using proteinase K treatment followed by thermal shock

PLOS ONE

Dear Dr. Paulo Vitor Marques Simas,

Thank you for submitting your manuscript to PLOS ONE. After careful consideration, we feel that it has merit but does not fully meet PLOS ONE’s publication criteria as it currently stands. Therefore, we invite you to submit a revised version of the manuscript that addresses the points raised during the review process.

We look forward to receiving your revised manuscript.

Kind regards,

Shawky M. Aboelhadid, PhD

Academic Editor

PLOS ONE

Reviewers' comments:

Reviewer's Responses to Questions

**Comments to the Author**

1. If the authors have adequately addressed your comments raised in a previous round of review and you feel that this manuscript is now acceptable for publication, you may indicate that here to bypass the “Comments to the Author” section, enter your conflict of interest statement in the “Confidential to Editor” section, and submit your "Accept" recommendation.

Reviewer #2: (No Response)

Reviewer #3: (No Response)

2. Is the manuscript technically sound, and do the data support the conclusions?

Reviewer #2: Yes

Reviewer #3: Yes

3. Has the statistical analysis been performed appropriately and rigorously? 

Reviewer #2: I Don't Know

Reviewer #3: I Don't Know

4. Have the authors made all data underlying the findings in their manuscript fully available?

Reviewer #2: Yes

Reviewer #3: No

5. Is the manuscript presented in an intelligible fashion and written in standard English?

Reviewer #2: Yes

Reviewer #3: Yes

6. Review Comments to the Author

Reviewer #2: (No Response)

Reviewer #3: I appreciate the author’s effors in rewriting the manuscript and re-plotting the data. The manuscript is greatly improved also regarding language and only minor points remain to be corrected, mainly related to phrasing.

Some minor typos/language errors appear in this version that were absent in the previous version, yet are not change tracked – I assume the authors are doing this manually. For future work, I suggest using the “Compare Documents” feature of MS Word or something similar (in Word: Review > Compare > Compare two versions of a document) – this way, changes are tracked automatically and completely.

Line 23: “during pandemic” > “during the pandemic”

Line 26 “diagnose the virus carrier” > “diagnose patients with suspected SARS-CoV-2 infection”

Line 35 “using a commercial kit with an alternative protocol” – but the alternative protocol exactly *doesn’t* use a commercial kit?

Line 42 “did not find significantly differ between them” – please rephrase, I assume you mean “did not differ significantly”

Line 50 “Thess viruses” > these viruses (This was also not change-tracked again, but appeared within the newest version while being correct in the previous one).

Line 189 “Was more significant” this implies a statistical test, if not done rephrase to “was higher”

Line 195: “p and F values represented a significant difference” – please explain what these values mean here

Line 222: “May be a challenge not accepted by other researchers” – sounds odd, rephrase or remove.

Line 224: “sucessful results” – rephrase, maybe “yielded promising results”

Line 246: “this was not denoted by us” – meaning your protocol didn’t have issues with low Ct? If yes, rephrase, e.g. “which was not observed in our study”

7. PLOS authors have the option to publish the peer review history of their article (what does this mean?). If published, this will include your full peer review and any attached files.

Reviewer #2: No

Reviewer #3: No

---

## [Author Response · Author response to Decision Letter 2]

1 Mar 2021

Response to the Editor and Reviewers

Please authors, you must be replied to the reviewer comments. The manuscript should be revised by expertise in English language. 

R: The authors carefully replied to all questions from the reviewers. In addition, the protocol is now available in https://doi.org/10.17504/protocols.io.br6bm9an. 

Responses or comments related to section “Comments to the Author”

1. If the authors have adequately addressed your comments raised in a previous round of review and you feel that this manuscript is now acceptable for publication, you may indicate that here to bypass the “Comments to the Author” section, enter your conflict-of-interest statement in the “Confidential to Editor” section, and submit your "Accept" recommendation.

Reviewer #2: (No Response)

Reviewer #3: (No Response)

Authors: No comment.

2. Is the manuscript technically sound, and do the data support the conclusions?

Reviewer #2: Yes

Reviewer #3: Yes

Authors: No comment.

3. Has the statistical analysis been performed appropriately and rigorously?

Reviewer #2: I Don't Know

Reviewer #3: I Don't Know

Authors: No comment.

4. Have the authors made all data underlying the findings in their manuscript fully available?

The PLOS Data policy requires authors to make all data underlying the findings described in their manuscript fully available without restriction, with rare exception (please refer to the Data Availability Statement in the manuscript PDF file). The data should be provided as part of the manuscript or its supporting information or deposited to a public repository. For example, in addition to summary statistics, the data points behind means, medians and variance measures should be available. If there are restrictions on publicly sharing data – e.g. participant privacy or use of data from a third party – those must be specified.

Reviewer #2: Yes

Reviewer #3: No

Authors: We made available all data, or in the manuscript or in the supplementary material. Nevertheless, if we must submit other information, please, let us know what material we must sharing, and we unquestionably will provide it instantly.

5. Is the manuscript presented in an intelligible fashion and written in standard English?

Reviewer #2: Yes

Reviewer #3: Yes

Authors: No comment.

6. Review Comments to the Author

Reviewer #2: (No Response)

Reviewer #3: I appreciate the author’s efforts in rewriting the manuscript and re-plotting the data. The manuscript is greatly improved also regarding language and only minor points remain to be corrected, mainly related to phrasing.

Some minor typos/language errors appear in this version that were absent in the previous version yet are not change tracked – I assume the authors are doing this manually. For future work, I suggest using the “Compare Documents” feature of MS Word or something similar (in Word: Review > Compare > Compare two versions of a document) – this way, changes are tracked automatically and completely.

Line 23: “during pandemic” > “during the pandemic”

Authors: This sentence was reformulated according suggested by the reviewer.

Line 26 “diagnose the virus carrier” > “diagnose patients with suspected SARS-CoV-2 infection”

Authors: This sentence was reformulated according suggested by the reviewer.

Line 35 “using a commercial kit with an alternative protocol” – but the alternative protocol exactly *doesn’t* use a commercial kit?

Authors: This sentence was reformulated to eliminate this misunderstanding, according suggested by the reviewer.

Line 42 “did not find significantly differ between them” – please rephrase, I assume you mean “did not differ significantly”

Authors: This sentence was reformulated to eliminate this misunderstanding, according suggested by the reviewer.

Line 50 “Thess viruses” > these viruses (This was also not change-tracked again, but appeared within the newest version while being correct in the previous one).

Authors: This sentence was reformulated according suggested by the reviewer.

Line 189 “Was more significant” this implies a statistical test, if not done rephrase to “was higher”

Authors: This sentence was reformulated according suggested by the reviewer.

Line 195: “p and F values represented a significant difference” – please explain what these values mean here. 

Authors: This sentence was reformulated adding this information, according suggested by the reviewer.

Line 222: “May be a challenge not accepted by other researchers” – sounds odd, rephrase or remove.

Authors: This sentence was deleted, according suggested by the reviewer.

Line 224: “sucessful results” – rephrase, maybe “yielded promising results”

Authors: This sentence was reformulated according suggested by the reviewer.

Line 246: “this was not denoted by us” – meaning your protocol didn’t have issues with low Ct? If yes, rephrase, e.g. “which was not observed in our study”

Authors: This sentence was reformulated according suggested by the reviewer.

7. PLOS authors have the option to publish the peer review history of their article (what does this mean?). If published, this will include your full peer review and any attached files.

Do you want your identity to be public for this peer review? For information about this choice, including consent withdrawal, please see our Privacy Policy.

Reviewer #2: No

Reviewer #3: No

Authors: No comment.

---

## [Editor Report · Decision Letter 3]

4 Mar 2021

PONE-D-20-28949R3

A faster and less costly alternative for RNA extraction of SARS-CoV-2 using proteinase K treatment followed by thermal shock

PLOS ONE

Dear Dr. Paulo Vitor ,

Thank you for submitting your manuscript to PLOS ONE. After careful consideration, we feel that it has merit but does not fully meet PLOS ONE’s publication criteria as it currently stands. Therefore, we invite you to submit a revised version of the manuscript that addresses the points raised during the review process.

ACADEMIC EDITOR:

Please, submit the correct version of the revised manuscript. 

We look forward to receiving your revised manuscript.

Kind regards,

Shawky M. Aboelhadid, PhD

Academic Editor

PLOS ONE
---

## [Author Response · Author response to Decision Letter 3]

4 Mar 2021

Response to the Editor and Reviewers

Please authors, you must be replied to the reviewer comments. The manuscript should be revised by expertise in English language. 

R: The authors carefully replied to all questions from the reviewers. In addition, the protocol is now available in https://doi.org/10.17504/protocols.io.br6bm9an. 

Responses or comments related to section “Comments to the Author”

1. If the authors have adequately addressed your comments raised in a previous round of review and you feel that this manuscript is now acceptable for publication, you may indicate that here to bypass the “Comments to the Author” section, enter your conflict-of-interest statement in the “Confidential to Editor” section, and submit your "Accept" recommendation.

Reviewer #2: (No Response)

Reviewer #3: (No Response)

Authors: No comment.

2. Is the manuscript technically sound, and do the data support the conclusions?

Reviewer #2: Yes

Reviewer #3: Yes

Authors: No comment.

3. Has the statistical analysis been performed appropriately and rigorously?

Reviewer #2: I Don't Know

Reviewer #3: I Don't Know

Authors: No comment.

4. Have the authors made all data underlying the findings in their manuscript fully available?

The PLOS Data policy requires authors to make all data underlying the findings described in their manuscript fully available without restriction, with rare exception (please refer to the Data Availability Statement in the manuscript PDF file). The data should be provided as part of the manuscript or its supporting information or deposited to a public repository. For example, in addition to summary statistics, the data points behind means, medians and variance measures should be available. If there are restrictions on publicly sharing data – e.g. participant privacy or use of data from a third party – those must be specified.

Reviewer #2: Yes

Reviewer #3: No

Authors: We made available all data, or in the manuscript or in the supplementary material. Nevertheless, if we must submit other information, please, let us know what material we must sharing, and we unquestionably will provide it instantly.

5. Is the manuscript presented in an intelligible fashion and written in standard English?

Reviewer #2: Yes

Reviewer #3: Yes

Authors: No comment.

6. Review Comments to the Author

Reviewer #2: (No Response)

Reviewer #3: I appreciate the author’s efforts in rewriting the manuscript and re-plotting the data. The manuscript is greatly improved also regarding language and only minor points remain to be corrected, mainly related to phrasing.

Some minor typos/language errors appear in this version that were absent in the previous version yet are not change tracked – I assume the authors are doing this manually. For future work, I suggest using the “Compare Documents” feature of MS Word or something similar (in Word: Review > Compare > Compare two versions of a document) – this way, changes are tracked automatically and completely.

Line 23: “during pandemic” > “during the pandemic”

Authors: This sentence was reformulated according suggested by the reviewer.

Line 26 “diagnose the virus carrier” > “diagnose patients with suspected SARS-CoV-2 infection”

Authors: This sentence was reformulated according suggested by the reviewer.

Line 35 “using a commercial kit with an alternative protocol” – but the alternative protocol exactly *doesn’t* use a commercial kit?

Authors: This sentence was reformulated to eliminate this misunderstanding, according suggested by the reviewer.

Line 42 “did not find significantly differ between them” – please rephrase, I assume you mean “did not differ significantly”

Authors: This sentence was reformulated to eliminate this misunderstanding, according suggested by the reviewer.

Line 50 “Thess viruses” > these viruses (This was also not change-tracked again, but appeared within the newest version while being correct in the previous one).

Authors: This sentence was reformulated according suggested by the reviewer.

Line 189 “Was more significant” this implies a statistical test, if not done rephrase to “was higher”

Authors: This sentence was reformulated according suggested by the reviewer.

Line 195: “p and F values represented a significant difference” – please explain what these values mean here. 

Authors: This sentence was reformulated adding this information, according suggested by the reviewer.

Line 222: “May be a challenge not accepted by other researchers” – sounds odd, rephrase or remove.

Authors: This sentence was deleted, according suggested by the reviewer.

Line 224: “sucessful results” – rephrase, maybe “yielded promising results”

Authors: This sentence was reformulated according suggested by the reviewer.

Line 246: “this was not denoted by us” – meaning your protocol didn’t have issues with low Ct? If yes, rephrase, e.g. “which was not observed in our study”

Authors: This sentence was reformulated according suggested by the reviewer.

7. PLOS authors have the option to publish the peer review history of their article (what does this mean?). If published, this will include your full peer review and any attached files.

Do you want your identity to be public for this peer review? For information about this choice, including consent withdrawal, please see our Privacy Policy.

Reviewer #2: No

Reviewer #3: No

Authors: No comment.

---

## [Editor Report · Decision Letter 4]

8 Mar 2021

A faster and less costly alternative for RNA extraction of SARS-CoV-2 using proteinase K treatment followed by thermal shock

PONE-D-20-28949R4

Dear Dr. Paulo Vitor Marques Simas,

We’re pleased to inform you that your manuscript has been judged scientifically suitable for publication and will be formally accepted for publication once it meets all outstanding technical requirements.

Kind regards,

Shawky M. Aboelhadid, PhD

Academic Editor

PLOS ONE
---

## [Editor Report · Acceptance letter]

16 Mar 2021

PONE-D-20-28949R4 

A faster and less costly alternative for RNA extraction of SARS-CoV-2 using Proteinase k treatment followed by thermal shock 

Dear Dr. Simas:

I'm pleased to inform you that your manuscript has been deemed suitable for publication in PLOS ONE. Congratulations! Your manuscript is now with our production department. 

Kind regards, 

on behalf of

Professor Shawky M. Aboelhadid 

Academic Editor

PLOS ONE